# Acute Promyelocytic Leukemia with Rare Genetic Aberrations: A Report of Three Cases

**DOI:** 10.3390/genes14010046

**Published:** 2022-12-23

**Authors:** Guang Liu, Lanting Liu, Daniel Di Bartolo, Katie Y. Li, Xia Li

**Affiliations:** 1Genetics/Genomics Division, Sonora Quest Laboratories, University of Arizona College of Medicine, Phoenix, AZ 85034, USA; 2AmeriPath Indiana, Indianapolis, IN 46219, USA; 3AmeriPath Northeast, Shelton, CT 06484, USA; 4Boston University School of Medicine and Boston Medical Center, Boston, MA 02118, USA; 5Department of Pathology, University of Arizona College of Medicine, Sonora Quest Laboratories, Phoenix, AZ 85034, USA

**Keywords:** acute promyelocytic leukemia, *PML::RARA*, variant *RARA* rearrangement, chromosome analysis, FISH, NGS, *FLT3*-ITD

## Abstract

Acute promyelocytic leukemia (APL) is a unique subtype of acute myeloid leukemia (AML) that is characterized by the *PML::RARA* fusion or, more rarely, a variant *RARA* translocation. While APL can be clinically suspected, diagnosis of APL requires genetic confirmation. Targeted therapy such as all-trans-retinoic acid (ATRA) and arsenic trioxide (ATO) has dramatically improved the prognosis of APL patients, but this is dependent on timely genetic testing as different fusions and/or mutations can affect therapeutic outcomes. Here we report three APL cases with various genetic aberrations: cryptic *PML::RARA* fusion, variant *RARA* rearrangement, and typical *PML::RARA* fusion with co-existing *FLT3-ITD* mutation. They serve to illustrate the utility of integrating genetic testing, using chromosome analysis, fluorescence in situ hybridization (FISH), reverse transcriptase-polymerase chain reaction (RT-PCR), and next-generation sequencing (NGS) in providing a detailed understanding of the genetic alterations underlying each patient’s disease.

## 1. Introduction

Acute promyelocytic leukemia (APL) is a subset of acute myeloid leukemia (AML) in which abnormal promyelocytes predominate. It accounts for 5–8% of AML cases. The disease can occur at any age but mainly affects middle-aged adults [1,2]. APL cells may contain Auer rods and kidney-shaped or bilobed nuclei, along with high-level myeloperoxidase expression. The genetic hallmark of APL is the fusion of the *RARA* gene at chromosome 17q21.2 with either the *PML* gene at chromosome 15q24.1 or, rarely, another partner gene. Other chromosomal aberrations, such as 7q deletion and trisomy 8, can also be present. Alterations in *FLT3,* such as an internal tandem duplication (*FLT3*-ITD), tyrosine kinase domain mutation (*FLT3-*TKD), and others, have been reported in APL cases [3,4,5]. APL immunophenotypic characteristics include high side scatter, forward scatter, negative expression of CD34 and HLA-DR, and positive expression of CD13, CD33, and CD117 [6,7]. An important distinguishing feature of APL is the frequent association with disseminated intravascular coagulation (DIC), leading to a high rate of mortality and morbidity if not diagnosed and treated immediately [8]. Targeted therapy with all-trans retinoic acid (ATRA) in conjunction with arsenic trioxide (ATO) has fortunately reduced the risk of hemorrhagic complications and transformed APL into a highly curable neoplasm. Therefore, the rapid and accurate establishment of APL diagnosis is critical for timely and effective treatment [9].

While APL can be suspected based on clinical presentation, cellular morphology, and immunophenotypic features, its diagnosis requires genetic confirmation of *PML::RARA* fusion or variant *RARA* rearrangement. Most Cytogenetics laboratories perform FISH with probes for *PML* and *RARA* for rapid detection of *PML::RARA* fusion, followed by FISH with break-apart probes for *RARA* if no fusion is identified, but rather three signals for *RARA* are evident. Chromosome analysis is also performed to evaluate for the (15;17) translocation [or a less common (v;17) translocation] associated with APL, and to identify any additional karyotypic complexity [10,11]. Reverse transcriptase polymerase chain reaction (RT-PCR), or real-time quantitative PCR, primarily used in subsequent analyses to assess disease status, can also be utilized on initial diagnosis if the cytogenetics testing is negative but clinical suspicion of APL is high. NGS assays with RNA-seq capability, such as the NGS Oncomine Myeloid Assay used in our lab, can detect *PML::RARA* fusion and identify variant *RARA* rearrangements as well as uncover other molecular alterations.

Here we report three cases of APL that demonstrate the utility of comprehensive laboratory testing in rendering an accurate diagnosis and in providing a better understanding of the genetic profile underlying each patient’s disease.


**Case 1**


The patient is a 32-year-old female with night sweats, pancytopenia, decreased platelets, and low fibrinogen (57 mg/dL). Laboratory data showed the following abnormal findings: WBC = 2535/µL; RBC = 2.7 million cells/µL; Hgb = 8.5 gm/dL; Hct = 24.1%; and platelets = 13 K/µL. Her blood smear consisted predominantly of normochromic, normocytic red blood cells with scattered nucleated red blood cells, mild anisocytosis, and decreased granulocytes. Abnormal promyelocytes were occasionally identified. Her platelets were markedly decreased. No fibrin strands or platelet clumps were observed.

Her bone marrow aspirate was cellular and paucispicular. The H&E- and Giemsa-stained particle sections were composed of blood clots. The H&E- and Giemsa-stained decalcified biopsy was adequate and markedly hypercellular. The cellularity was >95%, and the myeloid-to-erythroid ratio was indeterminate. Most of the cells were abnormal promyelocytes and represented >90% of nucleated cells. The cells had reniform nuclei and variable amounts of granulation. Auer rods and APL cells were observed. Granulocytic differentiation to the neutrophil stage was not significant. Monocytic differentiation to the monocyte stage was not evident. Erythropoiesis and megakaryopoiesis were decreased. No background dyspoiesis was evident. Lymphocytes were not increased. No granulomas were identified. Iron stains of the aspicular aspirate and decalcified biopsy showed no stainable reticuloendothelial iron stores.

Immunohistochemistry for CD34 showed no interstitial increase or clustering. Flow cytometry showed that the aspirate contained 84% blast cells with a dim CD45(+)/CD34(−)/CD117(+)/CD13(+)/CD33(+)[99%]/HLA-DR(−) and myeloperoxidase (99.3%) immunophenotype indicative of acute myeloid leukemia and suggestive of acute promyelocytic leukemia. Fluorescence in situ hybridization (FISH) analysis was performed on bone marrow aspirate with the *PML::RARA* dual-color dual-fusion probe set (MetaSystems, Medford, MA, USA). Two hundred nuclei were examined, the results were negative for *PML::RARA* rearrangement. Subsequent analysis with *RARA* break-apart probes (XL *RARA* BA; Metasystems) was also negative for *RARA* rearrangement. Chromosome analysis revealed a normal female karyotype (Figure 1A). NGS Oncomine Myeloid Assay (Thermo Fisher Scientific, Waltham, MA, USA) was also performed, which detected *PML::RARA* fusion and a *WT1* mutation (Figure 1B). RT-PCR was concurrently performed and revealed the long-form transcript of *PML::RARA* fusion at a transcript level of 1478.933 with *ABL1* as the control gene. This assay detects the short form (bcr3), the long form (bcr1), and the variant exon 6 (bcr2) *PML::RARA* transcripts (Quest Diagnostics, San Juan Capistrano, CA, USA). Based on the molecular test results, FISH was re-evaluated. Enhancing probe signal strength with Metafer 4 imaging software (Version 4.1.5; Meta-Systems) allowed for a fusion signal with a very faint signal for *RARA* (green) to be evident on the long arm of chromosome 15. These FISH results demonstrate that the *PML::RARA* fusion in this patient has resulted from a cytogenetically cryptic insertion of *RARA* into the *PML* locus on 15q (Figure 1C,D).


**Case 2**


A 51-year-old male presented to the clinic due to fatigue and easy bruising. A morphologic examination of the peripheral blood smear demonstrated some immature/blast-like cells. These cells were of predominantly intermediate to large size with oval to folded/convoluted nuclei, fine nuclear chromatin, small nucleoli, and scanty, finely granular cytoplasm. Examination of the bone marrow aspirate smears revealed adequate cellularity for analysis. Granulocytic precursors consisted almost entirely of blasts with virtually no neutrophils. Erythroid precursors were markedly decreased but showed no significant morphologic changes. A differential count showed 78% blasts, 0% promyelocytes, 2% myelocytes, 2% metamyelocytes, 7% granulocytes, 1% monocytes, 0% eosinophils, 0% basophils, 7% erythroid cells, 3% lymphocytes, and 0% plasma cells. The M:E ratio was 12.7. Megakaryocytes were decreased. Prussian blue stain revealed decreased iron staining. Flow cytometry demonstrated 79% blasts that were CD13(+), CD33(+), CD117(+), CD34(−), CD14(−), CD64(−) and HLA-DR(−). A diagnosis of AML with immunophenotypic features suggestive of APL was rendered.

FISH was performed on bone marrow with the *PML::RARA* dual-color dual-fusion probe set (Abbott Diagnostics, Abbott Park, IL, USA). Two hundred nuclei were examined, and all were negative for *PML::RARA* rearrangement. However, three signals for the *RARA* gene were evident in 173/200 (86.5%) of the cells scored (Figure 2A). FISH with *RARA* dual-color break-apart probes (Abbott Diagnostics, Abbott Park, IL) was subsequently performed, which demonstrated split signals, consistent with rearrangement of *RARA* in 177/200 (88.5%) of cells scored (Figure 2B). FISH with an AML panel (Abbott Diagnostics, Abbott Park, IL), comprised of probes for *RUNX1* (21q22.3) & *RUNX1T1* (8q22), *BCR* (22q11) & *ABL1* (9q34.12), *KMT2A* (11q23), and *CBFB* (16q22.1), was also performed. Two hundred nuclei were examined for each probe, and the results demonstrated three signals for *RUNX1T1* in 68/200 (34.0%) of the cells scored, suggesting a gain of all or part of chromosome 8 (Figure 2C). Chromosome analysis identified a translocation between the long arms of chromosomes 11 and 17 [i.e., t(11;17)(q23;q21)]. Two cells also displayed an additional copy of chromosome 8 (Figure 2D). RT-PCR was negative for *PML::RARA* fusion (Quest Diagnostics, San Juan Capistrano, CA, USA). NGS analysis with the Oncomine Myeloid Assay (Thermo Fisher Scientific, Waltham, MA, USA) detected *ZBTB16::RARA* fusion and additional mutations in *IDH2, TET2,* and *SRSF2* genes (Figure 2E).


**Case 3**


A 26-year-old male presented with a two-week history of fever and bleeding gums. The bone marrow biopsy was hypercellular marrow with left-shifted maturation in the myeloid lineage. The flow cytometry revealed a 77% blast population with dim CD45, bright CD33+, CD34+ (subset), HLA-DR(−), CD13+ (dim), CD117+, CD56+ (subset), and cytoplasmic MPO was positive. The morphology showed an increased nuclear cytoplasmic ratio and bilobed nuclei with sliding plate morphology. The clinicopathological findings were concerning for APL, and thus the clinical team initiated ATRA (all-trans-retinoic acid) treatment while awaiting genetic diagnosis.

FISH was performed on bone marrow aspirate using the *PML::RARA* dual-color dual-fusion probe set (Abbott Diagnostics, Abbott Park, IL), the results of which were positive for *PML::RARA* rearrangement in 187/200 (93.5%) of the cells scored (Figure 3A). FISH analysis with an AML panel for *RUNX1* (21q22.3 & *RUNX1T1* (8q22), *BCR* (22q11) & *ABL1* (9q34.12), *KMT2A* (11q23), and *CBFB* (16q22.1) (Abbott) identified three signals for *RUNX1T1*, suggesting gain of all or part of chromosome 8 in 172/200 (86.0%) of the cells scored (Figure 3B). RT-PCR was also positive for *PML::RARA* fusion (Quest Diagnostics, San Juan Capistrano, CA). Chromosome analysis detected the (15;17) translocation associated with APL, and trisomy 8 in 15 of the 20 cells examined (Figure 3C). The concurrent NGS Oncomine Myeloid Assay (Thermo Fisher Scientific, Waltham, MA) detected *PML::RARA* fusion and *FLT3*-ITD mutation with 35.30% allele frequency (Figure 3D). 

## 2. Discussion

When an APL diagnosis is clinically suspected, the affected patient should be managed as a medical emergency. Providing accurate genetic diagnosis as quickly as possible thus plays a critical role. In some cases, this can be complicated by rare genetic aberrations, such as cryptic *PML::RARA* rearrangements, variant *RARA* rearrangements, and/or co-existing targetable mutations such as *FLT3*. The cases presented in this study illustrate the importance of utilizing various genetic testing to ensure an accurate and comprehensive diagnosis of APL. A summary of genetic testing utilized in all three cases is shown in Table 1.

FISH is utilized to rapidly evaluate for the *PML::RARA* gene fusion or other *RARA* rearrangements. However, there are rare instances in which such rearrangements go undetected by this methodology as demonstrated in Case 1. Chromosome analysis provides a global assessment of the entire genome and therefore can identify translocations associated with APL. It also picks up secondary chromosomal alterations such as trisomy 8 and can identify clonal evolution, as in Cases 2 and 3. With this methodology, too, there are instances of rare alterations resulting in *PML::RARA* fusion or other *RARA* rearrangements that are not detectable at the resolution limits of this analysis. Such cytogenetically cryptic fusions are only detected by molecular techniques, such as RT-PCR or NGS. 

Case 1 illustrates this point. The initial FISH analysis with both *PML::RARA* dual fusion probes and *RARA* break-apart probes were negative. Chromosome analysis showed a normal karyotype. NGS and quantitative RT-PCR, however, identified the *PML::RARA* fusion. Re-evaluation of FISH, upon enhancing probe signal strength, identified a faint signal for *RARA* colocalizing with signal for *PML* on the long arm of one chromosome 15, thus confirming *PML::RARA* rearrangement due to a cytogenetically cryptic insertion of *RARA* into the *PML* locus. The oncogenic *PML::RARA* fusion more often occurs on the derivative chromosome 15, as seen in this case, and less commonly on the derivative chromosome 17 or other chromosomes [12]. Although one report by Zacarria et al. showed that the *PML::RARA* fusion on 17q may indicate a poor response to ATRA [13], APL patients with cryptic *PML::RARA* fusion generally respond very well to the targeted therapies (including ATRA and ATO) and share a favorable prognosis, the same as the APL patients with a typical *PML::RARA* fusion [14,15].

Some cryptic fusions are only detectable by molecular techniques such as RT-PCR or NGS, which offer much higher sensitivity at the RNA level. NGS can also detect DNA mutations such as the *WT1* mutation described in Case 1, which has been reported as a recurrent alteration in newly diagnosed APL, and can impact treatment decisions [16]. Therefore, Case 1 illustrates that whenever there is a strong clinical suspicion for APL but negative chromosome analysis and FISH results, it is important to use molecular techniques to ensure that a cryptic *PML::RARA* fusion is not missed.

Variant *RARA* translocations involving partner genes other than *PML* have been reported, including *ZBTB16* (previously called *PLZF*) at 11q23.2, *NUMA1* at 11q13.4*, NPM1* at 5q35.1, and *STAT5B* at 17q21.2 [17]. Cases with these variant translocations should be diagnosed as APL with a variant *RARA* translocation [1]. *ZBTB16::RARA* translocation, shown in Case 2, is the most common *RARA* variant translocation reported [18]. In this rearrangement, the *RARA* gene (17q21) is fused to the zinc finger and BTB domain containing protein 16 (*ZBTB16*, 11q23), which is also called the *PML* zinc finger protein (*PLZF*) gene [18]. Patients with *ZBTB16::RARA* fusion usually have the typical clinical presentation of APL, and incidence of disseminated intravascular coagulation. However, they often have some morphological variations, such as predominance of cells with regular nuclei, many granules, absence of Auer rods, or increased number of Pelgeroid neutrophils [18,19]. Variant APL with t(11;17)(q23;q21), resulting in *ZBTB16::RARA* fusion, has been reported as being resistant to ATRA. Therefore, identifying this rearrangement is very important [19]. The FISH strategy employed by most, if not all, labs does not identify a fusion partner. Chromosome analysis can often help elucidate fusion partners involved, as demonstrated in Case 2, as long as the alteration is not cytogenetically cryptic. RT-PCR using primers specific for *PML::RARA* cannot detect variant fusion partners, as shown in Case 2. Depending on the platform, NGS testing, however, can detect *RARA* rearrangement with numerous partner genes and multiple types of transcripts as well as identify concurrent mutations in other genes such as *IDH2, TET2,* and *SRSF2* genes in Case 2. Both *IDH2* and *SRSF2* have been described in the literature as recurrent secondary changes in APL with *ZBTB16::RARA* [20]. NGS provides comprehensive detection of mutations and variant fusion partners and should be used whenever there is a *RARA* rearrangement not involving *PML*.

**Table 1 genes-14-00046-t001:** Summary of genetic testing utilized in all three cases.

Case No.	Karyotyping	FISH	RT-PCR (*PML::RARA)*	NGS
**Case 1**	46,XX [20]	Negative initially; cryptic single fusion on 15q upon enhancement	Positive	*PML::RARA* fusion A muation in *WT1*
**Case 2**	46,XY,t(11;17)(q23;q21) [18]/47,idem,+8 [2]	Negative for *PML::RARA* fusion, Positive for *RARA* rearrangment	Negative	*ZBTB16::RARA* fusion Mutations in *IDH2*, *ET2* and *SRSF2*
**Case 3**	47,XY,+8,t(15;17)(q21;q21) [15]/46,XY [5]	Positive for *PML::RARA* fusion	Positive	*PML::RARA* fusion *FLT3*-ITD

The turnaround time (TAT) for NGS, RT-PCR, and chromosome analysis in many centers is generally more than 5 days. FISH results are often ready in 24 h. The latter is therefore heavily relied upon for timely confirmation of diagnosis and start to treatment.

When the FISH results are negative, NGS testing can be critical. Laboratories continue to work on reducing TAT for NGS. This can be of particularly importance for cases in which FISH and/or chromosome analysis are negative but clinicopathological suspicion for APL is high. NGS can serve as a rapid way of evaluating for an atypical *PML::RARA* fusion that was not picked up cytogenetically or in identifying the fusion partner in cases with variant *RARA* rearrangements.

Genetic testing including chromosome analysis, FISH, and NGS performed for Case 3 identified t(15;17) and its associated *PML::RARA* fusion. Chromosome analysis and FISH also identified the concurrent gain of chromosome 8, the most common secondary change in APL [1], while NGS detected *FLT3-ITD* mutation, which can help with risk stratification in patients with APL [21,22]. *FLT3-*ITD mutation with a high variant allele frequency (VAF), associated with *PML::RARA* gene fusion and trisomy 8, all of which represent a poor prognosis for APL, especially in young patients [23]. Mutations involving *FLT3*, including *FLT3*-ITD and *FLT3*-TKD mutations, occur in 30–40% of APL. Patients with *FLT3*-ITD mutation are more likely to have significantly higher white blood cell count at diagnosis, higher risk of induction deaths, and lower overall survival rates than those without this mutation [21,22,23,24]. A meta-analysis of 11 publications found that patients with the *FLT3*-ITD mutation had lower overall survival (OS) rates (RR = 0.59, *p* < 0.001) and a 1.70-fold higher likelihood of dying of any cause than those without this mutation [24]. One study reported that ATRA-ATO might abrogate the negative prognostic impact of *FLT3-ITD* [25]. In Case 3, the Mut/WT ratio is 0.563. According to Schlenk et al., an allelic ratio > 0.51 was associated with unfavorable relapse-free and overall survival [26]. This case serves to illustrate the utility of NGS testing in patients with APL, as detection of additional molecular alterations may help stratify risk more precisely and allow for a more individualized approach to treatment.

The treatment of patients with APL is different from those with other AML subtypes. Thanks to advances in the diagnosis and treatment of this disease, APL is currently considered the most curable adult leukemia. There are many treatment options for APL patients, including all-trans retinoic acid (ATRA), arsenic trioxide (ATO), anthracyclines, antimetabolites, stem cell transplantation, and clinical trials. Many studies have demonstrated that the combination of ATO and ATRA is superior to the former standard of care, which includes anthracyclines (daunorubicin and idarubicin), for patients with low-risk APL. For patients with high-risk APL, combinations of ATO, ATRA, and anthracyclines are widely used; the antimetabolite, e.g., cytarabine, can also be added to induction or consolidation regimens. Finally, for patients who have persistent minimal residual disease (MRD), stem cell transplantation and clinical trials are available (https://www.lls.org/treatment-acute-promyelocytic-leukemia, accessed on 1 October 2022). The use of a comprehensive genetic testing approach can help to accurately stratify patients’ risk and ensure that they receive the appropriate therapy.

These cases demonstrate the benefit of using a comprehensive genetic testing approach to identify cryptic *PML::RARA* fusions that may go undetected by conventional cytogenetics alone (i.e., chromosome analysis and FISH analysis), to characterize variant *RARA* rearrangements and uncover concurrent molecular alterations and secondary cytogenetic changes that can help guide treatment decisions and improve patient outcome. In addition, close follow-up and monitoring disease progression are recommended for these patients.

## Figures and Tables

**Figure 1 genes-14-00046-f001:**
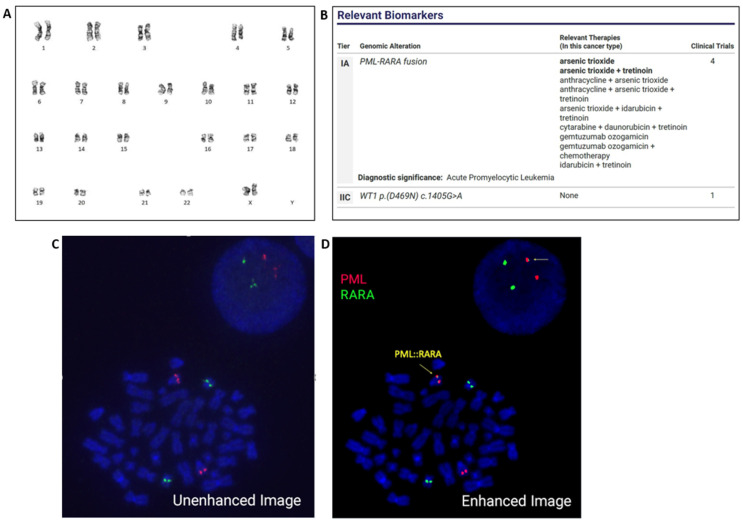
Chromosome analysis of the bone marrow revealed a normal female karyotype. ISCN: 46,XX (**A**). NGS detected *PML::RARA* fusion and a *WT1* mutation (**B**). Initial FISH analysis, which failed to identify the *PML::RARA* fusion (**C**). FISH re-evaluation, upon enhancing signal strength, identified a subtle cryptic insertion of *RARA* (green) into *PML* (red). Arrows indicate *PML::RARA* fusion at 15q (**D**).

**Figure 2 genes-14-00046-f002:**
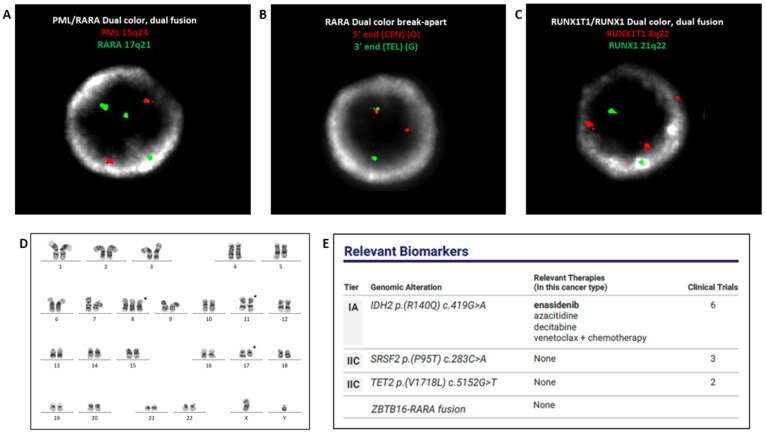
FISH with *PML::RARA* dual-color dual-fusion probes, negative for *PML::RARA* fusion, displaying three *RARA* (green) signals (**A**). FISH with *RARA* dual color break-apart probes displaying split signal, indicative of *RARA* rearrangement (**B**). FISH with the AML panel exhibiting three red signals for the pericentromeric region of chromosome 8 (**C**). Chromosome analysis identified gain of chromosome 8 and an (11;17) translocation. ISCN: 47,XY,+8, t(11;17)(q23;q21) (**D**). NGS analysis detected *ZBTB16::RARA* fusion and mutations in *IDH2, TET2,* and *SRSF2* genes (**E**).

**Figure 3 genes-14-00046-f003:**
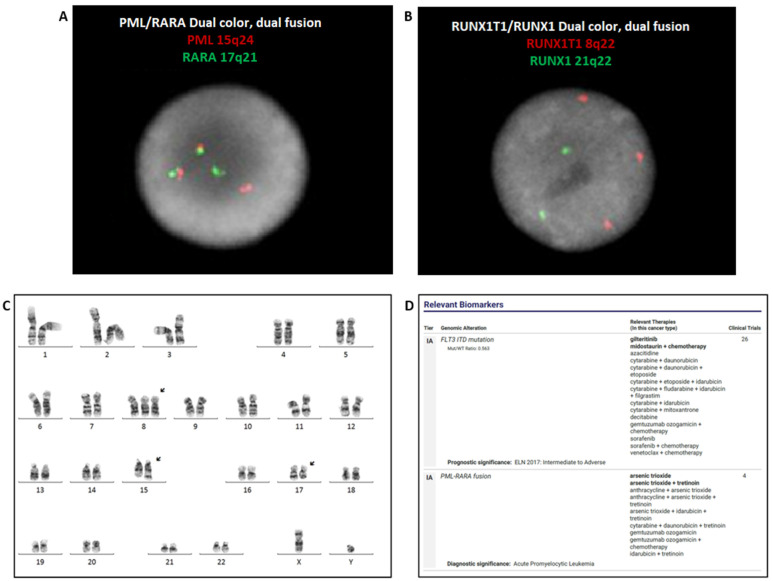
FISH with *PML::RARA* dual-color dual-fusion probes identified *PML::RARA* fusion (**A**). FISH with the AML panel displayed three *RUNX1T1* signals (red), suggesting a gain of chromosome 8 (**B**). Chromosome analysis showed a (15;17) translocation and concurrent gain of chromosome 8, consistent with the FISH analysis. ISCN: 47,XY,+8,t(15;17)(q21;q21) (**C**). NGS detected *PML::RARA* fusion and *FLT3*-ITD mutation (**D**).

## Data Availability

Exclude.

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
