# Peer review of "Acute Promyelocytic Leukemia with Rare Genetic Aberrations: A Report of Three Cases"

_genes, 2022, doi:10.3390/genes14010046_

Round 1

Reviewer 1 Report

Acute promyelocytic leukemia (APL) is a unique subtype of acute myeloid leukemia (AML). Although the dual target therapy is very successful and has dramatically improved the outcome in APL patients, the timely, accurate and effective diagnosis, combining clinical, MICM and multiple molecular techniques, is very important. Three APL cases carrying different rare genetic aberrations reported in the manuscript illustrate the importance of understanding and combination of multiple molecular techniques. 1、The purpose of diagnosis and clinical detection is optimal management of patients. But few treatment information was shown in the paper.  2、I wonder the primers of PML-RARA detection for the three cases are the same or not. If the result is positive, please provide the DNA electrophoresis results and Sanger sequence results, especially for Case 1. 3、The genetic aberrations are of different types, it is somewhat complicated to write them together.

Reviewer 2 Report

Thank you for the opportunity to review the manuscript titled ‘Acute Promyelocytic Leukemia with Rare Genetic Aberrations: A Report of Three Cases’ by Liu et al. The article is relevant and informative for readers and carries scientific merit. Please see some recommended changes for improvement:

1.     Line 34: Explain the frequency of FLT-3 ITD and TKD in APL

2.     Line 62: Include lab values supporting her presentation with differential of promyelocytes

3.     Line 145: Please enter the sub heading ‘Case 3’ so that the readers know that we have transitioned to the next case.

4.     Line 151: Please correct this: HLA-DR+ 9dim)

5.     Even though NGS, RT-PCR, cytogenetics testing are recommended, many centers have delay in getting these results back in a timely or consistent manner. Could the authors elaborate on what was the typical delay in getting some of these tests back and how did that effect the management of the patients mentioned in these cases? 

6.     Please have the discussion presented in a more organized manner. It is a little difficult to read in its current form. Please consider talking about conventional karyotyping together, PML-RARA FISH together and break-apart probes, RT-PCR together, and NGS together as opposed to moving back-and-forth between the different tests and different cases.

7.     Lines 248-250: please explain the survival rates as discussed 

8.     I would also recommend mentioning the clinical and prognostic implications of each aberration identified in the three cases in the discussion.
